



# Cross-validation of bias-corrected climate simulations is misleading

Douglas Maraun[1] and Martin Widmann[2]

[1]Wegener Center for Climate and Global Change, University of Graz, Brandhofgasse 5, 8010 Graz, Austria
[2]School of Geography, Earth and Environmental Sciences, University of Birmingham, Birmingham, B15 2TT, UK

*Correspondence to:* Douglas Maraun (douglas.maraun@uni-graz.at)

**Abstract.** We demonstrate both analytically and with a modelling example that cross-validation of free running bias-corrected climate change simulations against observations is misleading. The underlying reasoning is as follows: a cross-validation can have in principle two outcomes. A negative (in the sense of not rejecting a Null hypothesis), if the residual bias in the validation period after bias correction vanishes; and a positive, if the residual bias in the validation period after bias correction is large. It can be shown analytically that the residual bias depends solely on the difference between the simulated and observed change between calibration and validation period. These changes, however, depend mainly on the realisations of internal variability in the observations and climate model. As a consequence, also the outcome of a cross-validation is dominated by internal variability, and does not allow for any conclusion about the sensibility of a bias correction. In particular, a sensible bias correction may be rejected (false positive) and a non-sensible bias correction may be accepted (false negative). We therefore propose to avoid cross-validation when evaluating bias correction of free running bias-corrected climate change simulations against observations. Instead, one should evaluate temporal, spatial and process-based aspects.

*Copyright statement.* TEXT

## 1 Introduction

Bias correction is a widely-used approach to postprocess climate model simulations before they are applied in impact studies (e.g. Gangopadhyay et al., 2011; Hagemann et al., 2013; Girvetz et al., 2013; Warszawski et al., 2014). A wide range of different correction methods has been developed, ranging from simple additive or multiplicative corrections to quantile-based approaches. For reviews of bias correction see Teutschbein and Seibert (2012), Maraun (2016) and the book by Maraun and Widmann (2018).

The performance of a bias correction is typically evaluated against independent observational data, which have not entered the calibration of the correction function. For instance, Piani et al. (2010a), Piani et al. (2010b), Li et al. (2010) and Dosio and Paruolo (2011) apply the holdout method, i.e., they calibrate the method on a calibration period and evaluate it on a non-overlapping validation period. Some authors even apply a full cross-validation, most often by permuting calibration and validation period (a 2-fold cross validation Gudmundson et al., 2012).



Cross validation is a well known and widely used statistical concept to assess the skill of predictive statistical models (Stone, 1974; Efron and Gong, 1983). It has been successfully applied in the atmospheric sciences, e.g., in weather forecasting (Jolliffe and Stephenson, 2003; Mason, 2008; Wilks, 2006) and perfect predictor experiments of downscaling methods (Themeßl et al., 2011; Maraun et al., 2015).

In climate change applications, however, the setting is typically different from a weather forecasting or perfect predictor setting: here, the model is running free, i.e., only external forcings are common to observation and simulation. Internal climate variability on all scales is independent and not synchronised. In this setting, the aim is not to assess predictive power, e.g. on a day-by-day or season-by-season basis, as in weather forecasting - in fact, by construction it cannot be assessed. Importantly, also observed and simulated long-term trends may differ substantially, just because of different random realisations of long-
term modes of variability. Prominent examples of such modes are the Pacific Decadal Oscillation (PDO, Mantua et al., 1997) and the Atlantic Multidecadal Oscillation (AMO, Schlesinger and Ramankutty, 1994).

These differences have crucial implications for the application of cross-validation or any evaluation on independent data. Our results build upon a recent study by Maraun et al. (2017) who demonstrated that in a climate change setting cross-validation of marginal aspects is not able to identify bias correction skill. Here, we additionally show that the outcome of a cross-validation
is essentially random and independent of the sensibility of the bias correction. We will demonstrate these consequences for the holdout method, but they can of course be generalised to any type of cross-validation.

We will discuss the specific context of climate change simulations in Section 2. An analytical derivation of the cross-validation problem will be given in Section 3, a modelling example in Section 4. We will close with a discussion of the implications of our findings.

## 2   Cross Validation in the Climate Context

Cross validation has been developed to quantify the predictive skill of statistical models already in the 1930s, and has become widely used with the advent of modern computers (Stone, 1974; Efron and Gong, 1983). It has becomen a standard tool in weather and climate forecasting (Michaelsen, 1987; Jolliffe and Stephenson, 2003; Wilks, 2006; Mason, 2008).

The first major aim of cross validation is to eliminate artificial skill: if the statistical model is evaluated on the same data that
are used for calibration, the performance to predict new data will almost certainly be lower than the estimated skill. Hence, the model is calibrated only on a subset of the data, and evaluated on another - ideally independent - subset of the data. This so-called holdout method, however, uses each data point only either for calibration or validation and thus suffers from relatively high sampling errors.

The second major aim of cross validation is therefore to optimally use the data. To this end, the holdout method, i.e.,
training and validation, is repeated on different subsets of the data. The most simple approach is the so-called split sample method, where the data is just split once into two subsets. More advanced k-fold cross validation splits the data set into $k$ non-overlapping blocks; in each fold, $k-1$ blocks are used for calibration and the remaining block is used for validation.





In weather and climate predictions, the aim is to predict the weather, i.e. internal variability, with a given leadtime (say, 3 days or a season) at a desired timescale (say, 6 hours or a season). A typical evaluation assesses how well certain meteorological aspects are predicted: in weather forecasting, one may for instance be interested in the overall prediction accurracy, measured by the root-mean squared error between predicted and observed daily time-series. In a seasonal prediction, one may be interested

in the mean bias of the prediction, or in the predicted wet-day frequency over a season. In this context, a cross-validation makes perfect sense if the validation blocks are long compared to the prediction leadtime (and process memory).

Downscaling and bias correction methods are typically tested in perfect predictor or boundary condition experiments (Maraun et al., 2015), where predictors or boundary conditions are taken from reanalysis data. The aim of the downscaling in this context is not to predict internal variability ahead into the future, but rather to predict the local weather conditional on the state

of the large-scale weather (i.e., to simulate the correct local long-term weather statistics). Still, in such a setting cross validation makes perfect sense: the choice of reanalysis data as predictors/boundary conditions synchronises simulated and observed local variability on timescales beyond a few weeks, such that the evaluation framework is similar to the case of seasonal prediction.

In free running climate simulations, however, the situation is fundamentally different: here, any predictive power results only from external (e.g., anthropogenic) forcing at very long timescales, but internal variability is not synchronised at any timescale.

Yet long-term modes of internal climate variability, such as the Pacific decadal oscillation (PDO, Schlesinger and Ramankutty, 1994) and the Atlantic multidecadal oscillation (AMO, Schlesinger and Ramankutty, 1994), often mask forced climate trends even at multidecadal timescales (Deser et al., 2012; Maraun, 2013b). Thus, much of the difference between observed and simulated trends is not caused by model errors, but rather by random fluctuations of the climate system. This fact has strong implications for the evaluation of simulated trends (Bhend and Whetton, 2013; van Oldenborgh et al., 2013; Laprise, 2014),

but it is also the reason why cross-validation of bias correction fails in this context.

As any cross validation consists of repeat holdout evaluations, we will in the following only consider the holdout method. In Section 5 we will discuss how the following results generalise to a full cross validation.

## 3 Analytical Derivation

Consider a simulated time series $x_i$ and an observed time series $y_i$. Assume that an evaluation addresses the representation of

some statistic such as the long-term mean. Over the calibration period, we denote simulated and observed mean as $\bar{x}_{cal}$ and $\bar{y}_{cal}$, respectively. Correspondingly, we denote them as $\bar{x}_{val}$ and $\bar{y}_{val}$ over the validation period. Then an estimate for the bias over the calibration period is given as

$$\text{BIAS} = \bar{x}_{cal} - \bar{y}_{cal}. \tag{1}$$

Applying the bias estimate to the validation period, one obtains an estimate of the corrected mean over the validation period

$$\bar{x}_{val}^{corr} = \bar{x}_{val} - \text{BIAS} = \bar{x}_{val} - \bar{x}_{cal} + \bar{y}_{cal}. \tag{2}$$

The remaing residual bias is then

$$\text{BIAS}_{res} = \bar{x}_{val}^{corr} - \bar{y}_{val} = \bar{x}_{val} - \bar{x}_{cal} + \bar{y}_{cal} - \bar{y}_{val}. \tag{3}$$



This residual bias can be expressed in terms of the observed and simulated climate change signals. The change signal from calibration to validation period is defined as

$$\Delta x = \bar{x}_{val} - \bar{x}_{cal} \tag{4}$$

for the model and

$$\Delta y = \bar{y}_{val} - \bar{y}_{cal} \tag{5}$$

for the observations. Thus, the residul bias is given as

$$\mathrm{BIAS}_{res} = \Delta x - \Delta y. \tag{6}$$

For variables such as precipitation, one often considers relative changes. Here a corresponding derivation holds. The relative error is defined as

$$\mathrm{RE} = \bar{x}_{cal}/\bar{y}_{cal}, \tag{7}$$

and the corrected mean over the validation period is given as

$$\bar{x}_{val}^{corr} = \bar{x}_{val}/\mathrm{RE} = \bar{x}_{val} \cdot \bar{y}_{cal}/\bar{x}_{cal}. \tag{8}$$

The residual relative error results as

$$\mathrm{RE}_{res} = \bar{x}_{val}^{corr}/\bar{y}_{val} = \frac{\bar{x}_{val} \cdot \bar{y}_{cal}}{\bar{x}_{cal} \cdot \bar{y}_{val}}. \tag{9}$$

The relative change signal from calibration to validation period is defined as

$$\Delta x = \bar{x}_{val}/\bar{x}_{cal} \tag{10}$$

for the model and

$$\Delta y = \bar{y}_{val}/\bar{y}_{cal} \tag{11}$$

for the observations. Hence, the residul relative error is

$$\mathrm{RE}_{res} = \Delta x/\Delta y. \tag{12}$$

The residual bias or relative error could further be tested for significance, i.e., whether the bias corrected statistic $\bar{x}_{val}^{corr}$ is significantly different from the observed statistic $\bar{y}_{val}$ over the validation period. Thus, a holdout evaluation will yield a positive result (in the sense of rejecting the Null hypothesis, i.e., a non-zero residual bias) if the simulated change $\Delta x$ is different from the observed change $\Delta y$, and a negative result (i.e., a residual bias compatible with zero) if simulated and observed changes are indistinguishable.

Assume now that a given bias correction may or may not be sensible. Note in this context, that it is completely irrelevant to explicitly define what constitutes a sensible bias correction (but for a brief discussion see Section 4). Thus, in principle four cases are possible:





1. True negative: the bias correction is sensible, and the (bias corrected) climate model simulates a trend closely resembling the observed trend.

2. False positive: the bias correction is sensible, but due to internal climate variability, the (bias corrected) climate model simulates a trend different from the observed trend.

3. False negative: the bias correction is not sensible, but the (bias corrected) climate model for some reason simulates a trend similar to the observed trend. This case corresponds to the example given in Maraun et al. (2017).

4. True positive: the bias correction is not sensible, and the (bias corrected) climate model simulates a trend different from the observed trend.

The crucial point is: for typical record lengths, much of the difference between simulated and observed changes $\Delta x$ and $\Delta y$
will be caused by internal climate variability. Thus the result of a cross-validation, i.e., which of the four cases occurs, is purely random and does not say anything about the sensibility of the cross validation.

Maraun et al. (2017) considered case three: as the difference between simulated and observed trends on typical time-scalesno'of a few decades is dominated by internal variability, the holdout method is not suitable to identify a non-sensible bias correction. The reverse conclusion is that the holdout method - and consequently also a cross-validation - is not able to
corroborate whether a bias correction is sensible.

Yet the discussion above implies an even stronger conclusion: because case two might randomly occur, a sensible bias correction may be rejected by a cross validation. Thus, even more importantly, cross-validation results in the given context is not just useless, but even misleading.

## 4   Empirical Demonstration

To further illustrate the analytic findings, we will give examples of the four cases in an exaggerated modelling example. We consider mean summer (JJA) precipitation at four locations. As observational reference we select the E-OBS data set (Haylock et al., 2008). As calibration period we use 1956-1980, as evaluation period 1981-2005.

We need to select two examples where the given bias correction is sensible, and two where it is not. Finding a convincing example of a sensible bias correction has to rely on process understanding (Maraun et al., 2017). A major precondition is
that the climate model simulates a realistic present climate and a credible climate change (Maraun and Widmann, 2018). The former condition mainly involves a realistic representation of the large-scale circulation (Maraun et al., 2017). We therefore consider the following set up: as examples for a sensible bias correction, we consider summmer mean precipitation at two locations in Norway. Summer mean precipitation in Norway is dominated by large-scale precipitation, which can sensibly be assumed to be realistically simulated by current-generation general circulation models (GCMs). Specifically we choose a
transient simulation of the EC-EARTH (Hazeleger et al., 2010), a model which has been demonstrated to suffer from minor biases in the synoptic-scale atmospheric circulation over Europe only (Zappa et al., 2013). We assume that other potential problems such as mislocations (Maraun and Widmann, 2015) or scale gaps (Maraun, 2013a) are negligible for the considered



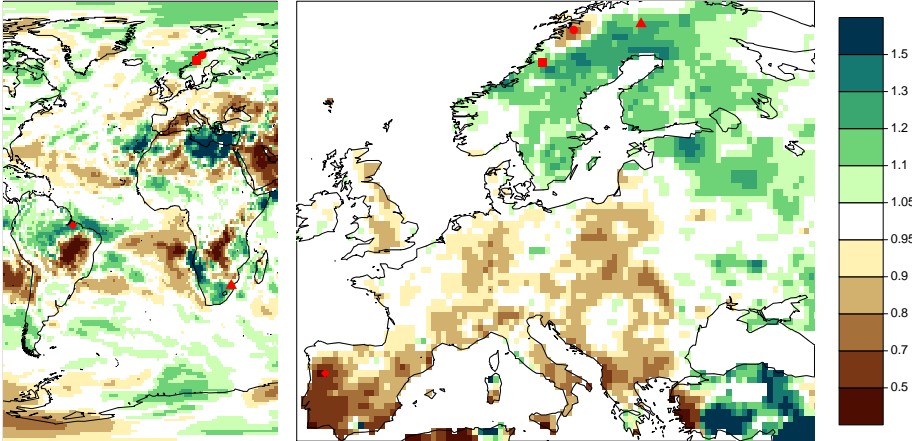

**Figure 1.** Maps of relative changes in boreal mean summer (JJA) precipitation, 1981-2005 relative to 1956-1980. Left: EC-EARTH, right: E-OBS. Square: case 1 (true negative); circle: case 2 (false positive); triangle: case 3 (false negative); diamond: case 4 (true negative).

locations and timescales. In this setting, we argue that a bias correction is in principle sensible. Two slightly different locations have been selected (Figure 1): in the Børgefjell region northeast of Trondheim observed and simulated trends are very similar (case 1). Further north, around the town of Bodø, the two trends are very different as observed precipitation has decreased whereas simulated precipitation has increased (case 2).

A discussion about the question when a bias correction makes no sense would go very much beyond the scope of this piece. Therefore, we follow the logic of Maraun et al. (2017) and select examples where model simulation and observation are taken from geographically far away and climatically rather different regions. The underlying idea is that for such cases, the model does not represent the target variable such that a bias correction is without doubt not sensible. Specifically, we consider the two following cases (see Figure 1): first, mapping simulated boreal summer mean precipitation from the sub-tropical Maputo area

(Mozambique, close to the South-African border) to the Taiga region of the Norwegian-Finnish boarder. Here, observed and simulated trends are randomly similar (case 3). Second, we map summer mean precipitation from the tropical climate at Belén in the Amazon delta to the mild and maritime climate of northern Portugal. Here, observed and simulated trends are randomly very different: positive in the Amazon delta, negative in Portugal (case 4).

    Figure 2 shows observed and simulated time series, the latter before and after bias correction, for the four cases we con-

sidered. Panels (a) and (b) show the sensible examples, panels (c) and (d) the non-sensible examples. In panels (a) and (c), observed and simulated trends randomly agree, in panels (b) and (d) they randomly disagree. As shown analytically in Section 3, the residual bias vanishes in cases (a) and (c) where the relative trends in observations and simulations are similar, and it does not vanish in (b) and (d), where the relative trends in observations and simulations disagree. The relevant cases are (b) and (c): in the former, the bias correction is in principle sensible, but the holdout method would suggest that it was not sensible

(false positive). In the latter, the bias correction is not sensible, but the holdout method would suggest that it was sensible (false





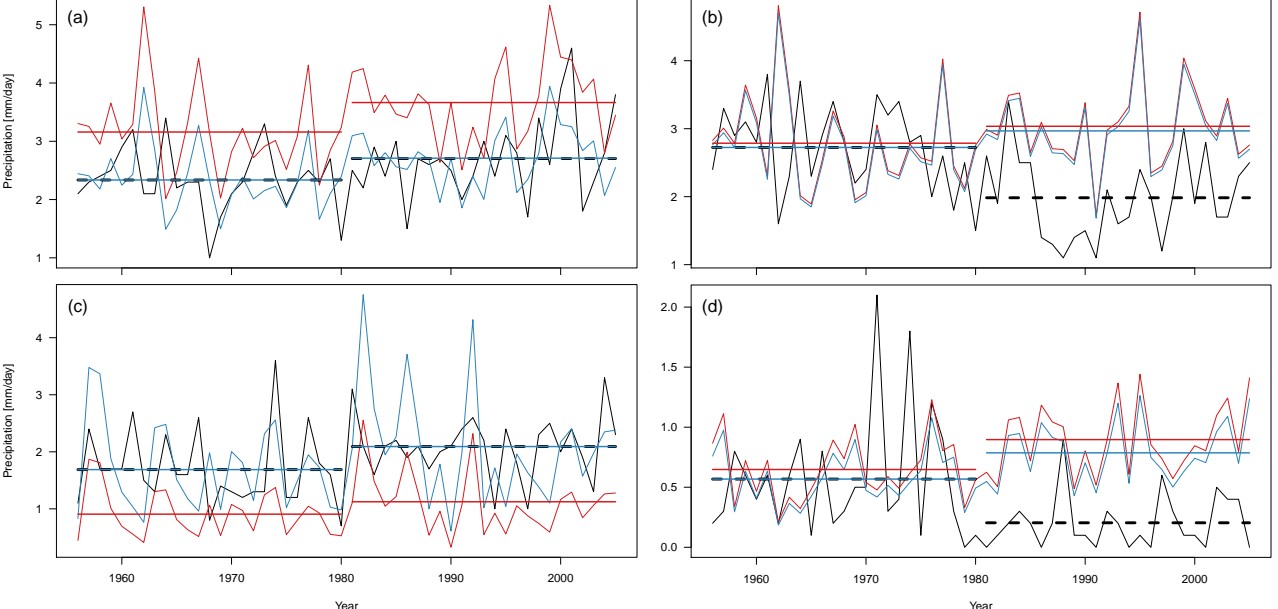

**Figure 2.** Time series of boreal summer (JJA) precipitation. (a) case 1 (true negative); (b) case 2 (false positive); (c) case 3 (false negative); (d) case 4 (true negative). Black: E-OBS; red: raw EC-EARTH; blue: bias-corrected EC-EARTH. The straight horizontal lines depict the long-term means over calibration and validation period respectively.

negative). These examples clearly illustrate our previous reasoning: the holdout method, and thus also cross-validation, yields misleading results when it is used to assess the sensibility of bias-corrected climate change simulations against observations.

## 5 Conclusions

We have demonstrated both analytically and with a modelling example that cross-validation of free-running bias-corrected climate change simulations against observations is misleading. The underlying reasoning is as follows: the result of a cross validation - a significant or non-significant residual bias in the validation period - depends on the difference between observed and simulated changes between calibration and validation period. These differences, however, depend mainly on the realisations of internal variability in the observations and climate model. These differences do not allow for any conclusion about the sensibility of a bias correction. As in any setting of significance testing, four cases are possible: true negative, false positive, false negative and true negative. The actual outcome in a given application is purely random.

We have derived these conclusions for the mean and the holdout method, where the bias correction is calibrated against one part of the data and validated against its complement. Yet the results can in principle be transferred to other statistics such as variances or individual quantiles, and to a full cross-validation. The residual mean bias, however, is always zero in a full cross-validation, as long as the individual folds have the same length. The reason is that changing calibration and validation period



changes the sign of the residual bias. When averaging the residual bias across the different folds it cancels out. For the variance or similar statistics, the outcome depends on the way the cross validation is carried out: if the residual bias is calculated for each fold separately and then averaged (as suggested in the classical literature), the behaviour is as for the mean. If the resdiual bias is calculated over a concatenated cross validated time series (as is typically done in the atmospheric sciences), the bias

5   correction in case (b) and (d) will yield extremely high residual biases (because the shift in the mean is not removed in the variance calculation).

The consequence of these findings is that cross-validation should not be used when evaluating bias correction of free-running climate simulations against observations. In fact, a framework for evaluating bias correction of climate simulations is still missing and not trivial. As discussed in Maraun et al. (2017), we propose to evaluate temporal, spatial and process-based

10  aspects of the simulated time series.

*Acknowledgements.* This study has been inspired by discussions in the EU COST Action ES1102 VALUE.





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
