# Peer review of "Cross-validation of bias-corrected climate simulations is misleading"

_Hydrology and Earth System Sciences, 2018_

## Referee Comment (RC1) · S. McGinnis (Referee) · 9 Jun 2018

This is a well-written manuscript that addresses an important point in the evaluation of bias-corrected climate simulations in a clear and thorough fashion. The argument is cogent, straightforward, and easy to understand. It covers an issue important to the readers of this journal, and I am pleased to recommend it for publication.

I can make only one comment of any substance, which is that I think it is a very slight exaggeration to say (page 5, lines 10-11, and again on page 7 line 10) that the result of cross-validation is *purely* random and says *nothing* about the sensibility. That would be the case if the difference between simulated and observed changes were caused entirely by internal variability and not merely dominated by it. Strictly speaking, the

result of the cross-validation is *almost entirely* random, and says *vanishingly little* about the sensibility of the cross-validation. However, this is a mere quibble, and does not change the conclusions; the manuscript is acceptable either way. I leave it up to the authors' judgment whether it will have greater impact on the reader to be scrupulously correct in every statement or to ever-so-slightly overstate the result to drive the point home.

I also have one very minor suggestion: the shapes of the red symbols in Figure 1 are difficult to make out at their current size. It might be beneficial to make the symbols somewhat larger or to give them a border in a contrasting color (e.g., black) to make it clearer which symbol is which.

Otherwise, I am reduced to noting a handful of typos and issues of grammar, as follows:

Page 1, line 7: move "also" after "is".

Page 2, line 9: move "also" after "may".

Page 2, line 21: change "has been" to "was", remove "already".

Page 2, line 29: move "optimally" after "use the data".

Page 3, line 1: "lead time" is two words.

Page 4, line 19: "residual" is misspelled.

Page 5, line 13: typo after "time-scales".

Page 7, line 6: replace hyphens with em-dashes.

Kudos to the authors for excellent work.

---

## Referee Comment (RC2) · U. Ehret (Referee) · 10 Jun 2018

'Cross-validation of bias-corrected climate simulations is misleading'

by D. Maraun and M. Widmann

Dear Editor, dear Authors,

I have reviewed the aforementioned work. My conclusions and comments are as follows:

**1. Scope**

The article is within the scope of HESS.

**2. Summary**

The authors start by giving a short overview on the scope and methods of bias correction (BC) as used in the field of climate model simulations. In this context they discuss the usage of cross validation to evaluate bias correction and distinguish the two cases 'perfect predictor setting' (where the boundary conditions of the problem are known) and 'climate change application' (where the free running model is not bounded by known boundary conditions). The authors argue that for the latter case, random realisations of long-term climate variability can render a classical split-sample cross-validation approach inapt to evaluate a BC method.

In section 3, the authors analyze a simple holdout cross-validation experiment for both an additive and multiplicative BC method: The BC value (factor) is determined in one partition of the available data and applied to the remaining data. The authors demonstrate that the residual bias in the validation data set, which is typically used as a measure of BC effectiveness, is sensitive to both the relative magnitudes of the simulated **and** observed change signals (changes between calibration and validation time). In short, climate variability between the calibration and validation period can lead to false conclusions about the effectiveness of a BC method.

In section 4, these findings are demonstrated at an exaggerated modelling example.

The authors conclude that cross validation should not be used when evaluating BC methods in free running climate simulations.

**3. Overall ranking**

The topic the authors tackle is highly relevant. The conclusions are convincingly supported by the analytical derivations shown in section 3. Therefore I think this is an important contribution to climate change research.

Nevertheless, I suggest the following **major** revisions:

- The analytical derivations in section 3 are straightforward, convincing and sufficient to support the author's arguments. I found the additional example in section 4 hard to understand. Also, as the example is constructed in an exaggerated manner to make the authors points clearer, I was not sure to which degree the conclusions based on this extreme case are transferable to 'normal' cases. Therefore I suggest deleting section 4 altogether. This will make the paper much clearer. If the authors wish to present an example, they can refer to the recent, excellent paper by the same authors (Maraun et al. 2017) where a similar example is presented. If section 4 is deleted, I further suggest to change the article from 'Research Article' to 'Technical Note', as it will be both short and dealing with a single, specific, technical question, which is what Technical Notes are meant for.

- I welcome the authors' discussion throughout the paper about when cross-validation approaches are valid and when not. What would be really helpful for the reader in the paper would be a short summary of the general problem by naming its main components and their interrelations (maybe

with a drawing) and strategies to decide whether cross-validation is appropriate for a given problem-setting or not. This could also serve as a sketch of the framework the authors mention in the last sentence of the conclusions. An incomplete list of the components:

- Length of available observational records [time]
- For a given observable (e.g. rainfall):
    - For a chosen spatial aggregation (e.g. 30x30 km grid): What is the aggregation time until (stationary) temporal variability has become statistically insignificant? [time]
    - For a chosen temporal aggregation (e.g. 1 year): What is the spatial aggregation until (stationary) spatial variability has become statistically insignificant [space]
- Existence and intensity of instationarities (trends): E.g. expressed by the time until a statistically significant change between the first and second half of a trend-afflicted time-series is detectable [time]
- Envisaged time of extrapolation beyond the observed period [time]
- By comparing the resulting timescales, it would be e.g. be possible to analyze:
    - With the available observational records, at which spatial/temporal aggregation can we separate instationarity from variability? This also puts a limit to the resolution of extrapolations and also allows to decide whether cross-validation makes sense or not.
- All of these components have already been mentioned by the authors in this or other papers, what I suggest here is to put them together in a compressed manner to frame the problem as a starting point for solution strategies.

- A last point: On page 6 line 5, the authors dismiss a discussion about the validity of BC as beyond the scope of the paper. However, when reading the paper, the natural question arising was 'If we do not have valid and agreed-upon methods to evaluate the effectiveness/appropriateness of a BC method in the observation period in the first place, how can we evaluate the validity of BC methods in the context of extrapolation, which is an even more involved problem?' So while I agree with the authors that an exhaustive discussion of this matter is impossible and beyond the scope of the paper, it should definitely be addressed here.

Yours sincerely,

Uwe Ehret

**References**

Maraun, D., G. Shepherd, T., Widmann, M., Zappa, G., Walton, D., Gutiérrez, J., Hagemann, S., Richter, I., Soares, P. M. M., Hall, A., and Mearns, L.: Towards process-informed bias correction of climate change simulations, nclimate3418 pp., 2017.

---

## Referee Comment (RC3) · S. McGinnis (Referee) · 11 Jun 2018

Could you say a bit more about what you found difficult about the example in section 4?

For me, the extended example was more intuitive and easier to apprehend than the analytical derivations in section 3, and I would much prefer that it remain in the final version of the paper. Perhaps there's something in the way that it's introduced that could stand to be clarified?
* * *

---

## Short Comment (SC1) · 11 Jun 2018

M. G. Grillakis

manolis@hydromech.gr

Authors argue that cross-validation of the free-running bias corrected climate model simulations is misleading. The main argument is that the remaining bias depends on the realizations of internal variability in the observations and climate model. Authors provide a good discussion about the limitations of cross validation in bias correction of free-running climate models.

To partly agree with the main point of the manuscript, cross-validation on independent data of free running climate models can become misleading. While this is not a well spread opinion it is not however a new concern. Drawbacks of split sample test in bias correction have been discussed in (Grillakis et al. 2017), where we mention that the

remaining bias of the validation period in split sample is a function of (i) the bias correction methodology's deficiency and (ii) the climate model deficiency itself to describe the validation period's climate, in aspects that are not intended to be bias corrected (i.e. long term modes of variability).

To further analyze the effect of "internal variability" on which authors attribute the remaining biases, it actually splits into two different and well defined reasons:

a) The first is how well synchronized to the observations can a free-running model be, in terms of multiyear modes of variability (such as PDO AMO etc). This is mainly random in a free-running model, as authors also discuss in the manuscript. However, it would be mentioned here that even a "perfect" model would not synchronize to the observations due to imperfections in the initial conditions, spin up effect, etc.

b) The second is that climate models are not able to precisely reproduce statistically these large scale modes as in reality they are not perfect. In that case, the remaining bias is related to the deficiency of the climate model to reproduce these multiyear persistencies, which would affect the results of cross validation, even in the case of a "synchronized" to the observations climate model run.

In (a), and by using a large period of data (times larger PDO AMO etc modes), cross validation will work well for calibration on the odd and validation on the even years (see Minville et al. 2014). This cross validation type would cancel out the synchronization issue of a "perfect" model.

In (b) the cross validation (again, using a large period of data) will reveal the weakness of the bias correction methodology to adjust the effect of the multiyear modes of variability. Considering that in typical bias correction applications, where ∼30 years of historical data are used for calibration to correct ∼100years of precipitation ahead (Grillakis et al. 2017), this is something that we expect from a bias correction method.

To summarize the above and before condemning the cross-validation in bias correction,

two more questions should be answered:

a) Is the cross-validation misleading regardless the length of the calibration- validation periods and the type of the holdout method?

b) Is the cross-validation inadequate to reveal the weaknesses of the bias correction method to adjust multiyear modes' effect on precipitation?

Other comments: • Authors do not refer to the version of EOBS data. Older versions of the dataset exhibit "no data" periods in the region of Turkey, that may be the source of the increased relative changes in Figure 1b. Also the EOBs dataset should be acknowledged according to the terms of use (http://surfobs.climate.copernicus.eu/dataaccess/access_eobs.php#datafiles) • P6-L16: Figure 2 shows averages, so do you mean "simulated average"?

References

Grillakis MG, Koutroulis AG, Daliakopoulos IN, Tsanis IK (2017) A method to preserve trends in quantile mapping bias correction of climate modeled temperature. Earth Syst Dyn Discuss 1–26. doi: 10.5194/esd-2017-53

Minville M, Cartier D, Guay C, et al (2014) Improving process representation in conceptual hydrological model calibration using climate simulations. Water Resour Res 50:5044–5073. doi: 10.1002/2013WR013857

---

## Referee Comment (RC4) · U. Ehret (Referee) · 12 Jun 2018

Dear Seth,

The reason I suggest to omit section 4 ist that the point the authors want to make is very clear from the analytical part in section 3. So for me there is no requirement to add the examples. As mentioned, one difficulty for me when interpreting the examples was how to transfer the magnitude of the effects from these extreme examples (extreme wrt to the non-sensibility of the second example) to typical application examples.

But: Omitting section 4 is a suggestion, not a hard requirement from my side. So, as you (and maybe also other readers) obviously benefitted from that section, I leave the decision up to the Editor.

[Figure]

Regards, Uwe

---

## Author Comment (AC1) · 11 Jul 2018

We very much appreciate the positive comments. We agree with the comment that the statement that one cannot learn anything from a cross validation in the setup discussed is too strong. We will therefore rephrase this statement and tune it down accordingly. In particular we will state that this depends on the length of the time series in comparison to the typical periodicity and aplitude of the dominant modes of variability (see comment by reviewer 2).

We will play around with some better colors/symbols for the red symbols. The issue is that they should be small not to mask the colors below. One solution could be using different colors instead of using different symbols.

[Figure]

We will include all minor comments listed.

---

## Author Comment (AC2) · 11 Jul 2018

We would like to thank the reviewer for the thoughtful and constructive comments.

Regarding the possible deletion of section 4, we strongly support the reasoning of reviewer 1 to keep the example as is. The reason is twofold:

1. we believe that the analytical derivation might be helpful for some readers, whereas others may prefer a strong illustrative example. This holds in particular as - to our experience - the role of internal variability in climate research is still often underestimated. The analytical derivation might then be dismissed as being purely academic reasoning without practical relevance.

2. choosing the Maraun et al. (2017) example would not suffice. In fact, the reason for

this short article was that - during the review process of the Maraun et al. (2017) paper we realised that the situation was even worse than laid out in that paper. There, the key starting point for the discussion was that cross validation may not be able to identify a nonsense bias correction. This is the false negative case in the manuscript at hand. Here we show additionally that there is another problematic case: that a sensible bias correction may be rejected (false positive). This case is not contained in the Maraun et al. (2017) example. Of course, it makes sense to furthermore show the true positive and true negative cases as well. We also do not believe that the exaggerated examples are limited in applicability/transferability to more realistic cases: the true negative case is a realistic case where a well performing climate model is successfully bias corrected - here, the case is not at all exaggerated. Similarly the false positive case, where the bias correction is sensible, but the residual bias does not vanish, is far from exaggerated: this is exactly the case we would like to highlight with this paper.

The other two cases are chosen to display wrong applications of bias correction in a convincing case. To avoid any discussion about the sensibility of bias correction in one or the other situation, we decided to take examples where anybody would agree. In fact, the false negative case - the correction does not make sense, but it is not rejected - is of a very similar character as the example chosen in Maraun et al. (2017). One may actually argue that the correction of temperature against precipitation from two different regions in that paper is even more exaggerated than the example here (where the same variables are chosen, but different locations). In a real application, of course, the problem will not be as obvious as constructed in our examples. Here, the user of BC has to carefully assess whether the bias correction makes sense at all (see discussion below). This discussion, however, is not the main focus of our manuscript.

To summarise, we suggest to keep the example as is. But we agree that a brief discussion on the transferability of our examples might help.

Regarding the suggestion of a general discussion of the contexts in which cross validation may make sense: we believe this would go well beyond the scope of our

manuscript. As indicated, this even depends on the way cross validation is carried out (in the "statisticians way" or in the "atmospheric scientists way"), with several subtleties. A thorough discussion could easily distract the reader from our main point.

We do agree, however, that it would be useful to add a discussion about the role of the length of the observational record in comparison with the periodicities and amplitude of the relevant mode(s) of variability. We will therefore add a brief discussion accordingly and will try - as suggested by the reviewer - to pull the different points together to highlight the overall issue in a concise manner.

Regarding the question how BC can be evaluated in the context of extrapolation: the reviewer is of course right that this is in general an open question. We have had a discussion of this issue in the Maraun et al. (2017) paper, where we highlight the fact that BC has to be accompanied by a thorough evaluation of non-corrected features (in particular temporal and spatial), by a process-based evaluation of the underlying climate model (in terms of location biases, relevant feedbacks etc.), by an assessment of potential artefacts, and by reasoning about representativeness and trend modifications. We will consider to add a summary of this discussion in the conclusions with a reference to the other paper.

---

## Author Comment (AC3) · 11 Jul 2018

We would like to thank Dr. Grillakis to direct us to his discussion paper. We will consider if and where to best refer to it in our manuscript. He raises an interesting point of biases in the representation of internal variability. We will consider whether this issue is relevant in the context of our paper.

Of course, he is right that even a model run with "perfect" boundaries will not be perfectly synchronised, but in many cases this effect should be minor on climatic time scales (e.g., Maraun & Widmann, HESS; 2015). Of course, this depends on the specific setup and on the correction method (e.g., a quantile mapping correction of extremes will need more data than a mean bias correction). But in RCMs where sea surface

temperatures are taken from observations and spin-up effects mainly result from the soil moisture initialisation, the randomness should be negligible when averaging over a 30 year period.

We do not fully agree with the comment on using odd and even years in the cross validation. Here, of course, the effect of long-term modes of variability would cancel out, but randomness due to interannual variations might still be a dominating effect.

Finally, the issue of length of the calibration/validation period will be discussed. The issue of whether cross-validation is inadequate to reveal the weaknesses of the bias correction method to adjust multiyear modes' effect on precipitation is a subtle one which goes far beyond out manuscript: in Maraun et al., Nat. Clim. Change (2017) we argue that often the question is not so much about the bias correction method, but rather about the skill of the underlying climate model.

---

## Author Response (AR1)

**Response to Reviewer Comments**

We would like to thank the reviewers for their helpful and supportive comments. Please find our point-by-point response below.

**Reviewer 1 (Seth McGinnis)**

> *I can make only one comment of any substance, which is that I think it is a very slight exaggeration to say (page 5, lines 10-11, and again on page 7 line 10) that the result of cross-validation is \*purely\* random and says \*nothing\* about the sensibility. That would be the case if the difference between simulated and observed changes were caused entirely by internal variability and not merely dominated by it. Strictly speaking, the result of the cross-validation is \*almost entirely\* random, and says \*vanishingly little\* about the sensibility of the cross-validation.*

We agree with the reviewer. Therefore we have adjusted the text as suggested (we used mostly instead of almost entirely though).

> *the shapes of the red symbols in Figure 1 are difficult to make out at their current size. It might be beneficial to make the symbols somewhat larger or to give them a border in a contrasting color (e.g., black) to make it clearer which symbol is which.*

We have added a border and slightly increased the size.

> *Page 1, line 7: move "also" after "is".*

We have kept the also, as the emphasis is slightly different (both versions are grammatically correct).

> *Page 2, line 9: move "also" after "may".*

Same as in the case before.

> *Page 2, line 21: change "has been" to "was", remove "already".*

Changed.

> *Page 2, line 29: move "optimally" after "use the data".*

Changed.

> *Page 3, line 1: "lead time" is two words.*

Changed.

> *Page 4, line 19: "residual" is misspelled.*

Changed.

> *Page 5, line 13: typo after "time-scales".*

Changed.

> *Page 7, line 6: replace hyphens with em-dashes.*

Changed.

**Reviewer 2 (Uwe Ehret)**

> *The analytical derivations in section 3 are straightforward, convincing and sufficient to support the author's arguments. I found the additional example in section 4 hard to understand.*
> *Also, as the example is constructed in an exaggerated manner to make the authors points clearer, I was not sure to which degree the conclusions based on this extreme case are transferable to 'normal' cases. Therefore I suggest deleting section 4 altogether. This will make the paper much clearer. If the authors wish to present an example, they can refer to the recent, excellent paper by the same authors (Maraun et al. 2017) where a similar example is presented.*

Regarding the possible deletion of section 4, we strongly support the reasoning of reviewer 1 to keep the example as is. The reason is twofold:
1. we believe that the analytical derivation might be helpful for some readers, whereas others may prefer a strong illustrative example. This holds in particular as - to our experience - the role of internal variability in climate research is still often underestimated. The analytical derivation might then be dismissed as being purely academic reasoning without practical relevance.
2. choosing the Maraun et al. (2017) example would not suffice. In fact, the reason for this short article was that - during the review process of the Maraun et al. (2017) paper we realised that the situation was even worse than laid out in that paper. There, the key starting point for the discussion was that cross validation may not be able to identify a nonsense bias correction. This is the false negative case in the manuscript at hand. Here we show additionally that there is another problematic case: that a sensible bias correction may be rejected (false positive). This case is not contained in the Maraun et al. (2017) example. Of course, it makes sense to furthermore show the true positive and true negative cases as well.
We also do not believe that the exaggerated examples are limited in applicability/transferability to more realistic cases: the true negative case is a realistic case where a well performing climate model is successfully bias corrected - here, the case is not at all exaggerated. Similarly the false positive case, where the bias correction is sensible, but the residual bias does not vanish, is far from exaggerated: this is exactly the case we would like to highlight with this paper.

The other two cases are chosen to display wrong applications of bias correction in a convincing case. To avoid any discussion about the sensibility of bias correction in one or the other situation, we decided to take examples where anybody would agree. In fact, the false negative case - the correction does not make sense, but it is not rejected - is of a very similar character as the example chosen in Maraun et al. (2017). One may actually argue that the correction of temperature against precipitation from two different regions in that paper is even more exaggerated than the example here (where the same variables are chosen, but different locations). In a real application, of course, the problem will not be as obvious as constructed in our examples. Here, the user of BC has to carefully assess whether the bias correction makes sense at all. This discussion, however, is not the main focus of our manuscript.

Therefore we kept the examples in the text. We added, however, some explanations and brief discussions on the transferability of these examples to real applications (page 6, line 11-15; page 7, line 5-8).

> *I welcome the authors' discussion throughout the paper about when cross-validation approaches are valid and when not. What would be really helpful for the reader in the paper would be a short summary of the general problem by naming its main components and their interrelations (maybe with a drawing) and strategies to decide whether cross-validation is appropriate for a given problem-setting or not. This could also serve as a sketch of the framework the authors mention in the last sentence of the conclusions.*

We believe a specific discussion of the contexts in which cross validation may make sense (including a figure) would go well beyond the scope of our manuscript. As indicated, this even depends on the way cross validation is carried out (in the "statisticians way" or in the "atmospheric

scientists way"), with several subtleties. A thorough discussion could easily distract the reader from our main point. We added, however, a short and rather general discussion of the issues influencing the sensibility of a cross validation. This discussion covers all the issues raised by the reviewer, but in a general way only (page 8, line 13-22).

> *On page 6 line 5, the authors dismiss a discussion about the validity of BC as beyond the scope of the paper. However, when reading the paper, the natural question arising was 'If we do not have valid and agreed-upon methods to evaluate the effectiveness/appropriateness of a BC method in the observation period in the first place, how can we evaluate the validity of BC methods in the context of extrapolation, which is an even more involved problem?' So while I agree with the authors that an exhaustive discussion of this matter is impossible and beyond the scope of the paper, it should definitely be addressed here.*

The reviewer is of course right that this is an entirely open question. We have had a discussion if this issue in the Maraun et al. (2017) paper, where we highlight the fact that BC has to be accompanied by a thorough evaluation of non-corrected features (in particular temporal and spatial), by a process-based evaluation of the underlying climate model (in terms of location biases, relevant feedbacks etc.), and by reasoning about representativeness and trend modifications. In the conclusions, we had a very short discussion of this topic. We have slightly extended this discussion to accommodate for the reviewer's comment (page 9).